# Effect of Both the Phase Composition and Modification Methods on Structural-Adsorption Parameters of Dispersed Silicas

**Tatyana Rakitskaya [1],\*, Tatyana Kiose [1], Kristina Golubchik [1], Viacheslav Baumer [2] and Vitaliya Volkova [1]**

[1]   Faculty of Chemistry and Pharmacy, Odessa I.I. Mechnikov National University, 2, Dvoryanska Street, 65082 Odessa, Ukraine; kiosetatyana@gmail.com (T.K.); golubchikko@gmail.com (K.G.); vita_volkova@rambler.ru (V.V.)

[2]   Institute of Monocrystals of National Academy of Sciences of Ukraine, 60, Nauki Avenue, 61072 Kharkiv, Ukraine; baumer@xray.isc.kharkov.com

\*   Correspondence: tlr@onu.edu.ua; Tel.: +38-067-559-3847

**Abstract:** Tripoli from two Ukrainian deposits was studied in its natural and modified forms. The investigation of natural and modified tripoli involves the identification of their phase compositions through X-ray diffraction and the analysis of their water vapor adsorption-desorption isotherms. The obtained results are evidence of changes in the structural-adsorption parameters of tripoli as a result of modification. Their treatment in boiling water or acid causes apparent alterations of contents of the main phases and sizes of their crystallites, whereas their calcination causes not only the dehydroxylation of surfaces and the agglomeration of phases, but even phase transformation in the case of carbonate tripoli. After analyzing water vapor adsorption-desorption isotherms of natural and modified tripolis, some correlations between their adsorption parameters, phase compositions, main phase contents and crystallite sizes have been found.

**Keywords:** Ukrainian tripolis; modification; structural-adsorption parameters; XRD method; water vapor adsorption-desorption isotherms

## 1. Introduction

Water vapor adsorption-desorption is one of the most important properties of natural and synthetic sorbents, determining possibilities for their use in ecological, energy-saving and nuclear safety applications [1–4]. The development and use of supported metallic, metal oxide, and metal-complex catalysts for air purification from toxic gaseous substances such as $PH_3$, $CO$, $SO_2$, and $O_3$ [5–8] is also connected with investigations of the adsorption-desorption parameters of the sorbents used as supports with respect to water vapor [5,6]. Natural sorbents, particularly dispersed silicas (tripolis, diatomites, etc.), are most acceptable for this. Natural sorbents are also suitable for the adsorption of heavy metals [9–12] and organic dyes [13,14], as well as being part of compositions for the removal of sulfur dioxide from air [15,16] and so on, due to their availability and low prices.

For obtaining the anchored metal complex catalysts, it is necessary to study not only the adsorption properties of sorbents towards metal ions [17] but also the affinity of water molecules to sorbent surfaces [5,6,18,19]. This is due to the fact that the activity of the anchored metal complex catalysts substantially depends on the water contents that are in them. Water is not removed from the catalysts after drying at 110 °c in the course of their preparation. Moreover, water is able to build up in the process of the catalyst operation in personal protective equipment, such as in gas masks or respirators.

This is due to the impossibility of keeping the air humidity constant. However, in many cases, some previous treatment (modification) of natural sorbents (such as boiling in water or acid or through calcination) is required for increasing the activity of the metal complex compounds that are anchored on them [20]. Those kinds of treatment result in changes in the structural-adsorption parameters and the physicochemical properties of supports, particularly in their water vapor adsorption capacities. These changes are well-examined in supports such as clinoptilolites and bentonites [21,22], however they are quite unexplored in the case of tripolis. Natural tripolis of different origins differ in their phase compositions and main phase ratios, and these parameters undoubtedly would change in their modified forms.

## 2. Materials and Methods

In this work, samples of natural tripoli from two Ukranian deposits were used: The Konopliansky deposit (denoted as N-Tr(K-I) and N-Tr(K-II)), where samples were mined at different depths of occurrence, and the Mohyliv-Podils'kyy deposit (denoted as N-Tr(MP)).

The samples that were obtained by boiling in distilled water for 1 h are denoted as $H_2O$-Tr(K-I), $H_2O$-Tr(K-II) and $H_2O$-Tr(MP).

The samples that were obtained by boiling in a 3 M $HNO_3$ solution for 1 h are denoted as 3H-Tr(K-I), 3H-Tr(K-II) and 3H-Tr(MP).

The samples that were obtained by calcination in air at 1000 °C for 1 h are denoted as 1000-Tr(K-I), 1000-Tr(K-II) and 1000-Tr(MP).

The samples were investigated using a Siemens D500 powder diffractometer (CuK$_\alpha$ radiation, $\lambda$ = 1.54178 Å) with a secondary beam graphite monochromator (Siemens AG, Munich, Germany). After thorough grinding with a pestle, each sample was placed into a glass cell of an enclosed volume of $2 \times 1 \times 0.1$ cm$^3$ for XRD pattern recording in the 2θ range from 0° to 90°, with a step of 0.03° and an accumulation time every 60 s.

Water vapor sorption was studied using a temperature-controlled 21 °C vacuum setup with a conventional McBain-Bakr silica-spring balance (Odessa I.I. Mechnikov National University, Odessa, Ukraine). Previously, the samples were degassed at 110 °C for 2 h. The error of measurement was ±2%. The specific surface areas were estimated with the Brunauer-Emmett-Teller (BET) method. Their pore size distribution curves were determined using desorption branches of their isotherms, and their pore radii were estimated using the Kelvin equation [23].

## 3. Results

### 3.1. Phase Compositions

Figure 1 shows some examples of the most differing XRD patterns for natural and thermally modified tripoli samples.

All samples of natural tripoli are crystalline, however, slightly amorphized. The maximum differences are observed in the 2θ region from 20° to 31°. For all natural tripolis, complex bands consisting of three peaks were observed in the 2θ range from 20° to 23°. These three peaks are characteristic of α-tridimite (α-trid), α-quartz, and β-cristobalite (β-crist). The most intense peak for N-Tr(K-II), belonging to the α-quartz phase, was located at 2θ = 26.600°. In contrast with this, the second α-quartz reflection for N-Tr(MP) was weak, whereas its most intense reflection was at 2θ = 29.359°, and an interplanar spacing where $d_{422}$ = 3.039 Å belongs to the calcite phase, indicating that N-Tr(MP) is a carbonate form of tripoli.

The calcite phase also demonstrates several more reflections of different intensities at 22.999, 35.931, 39.363, 43.127, 47.466 and 48.468°. All XRD patterns were treated with the Rietveld method. The results of the phase identification, phase contents, and crystallite sizes (D) of natural and modified tripolis are summarized in Table 1.

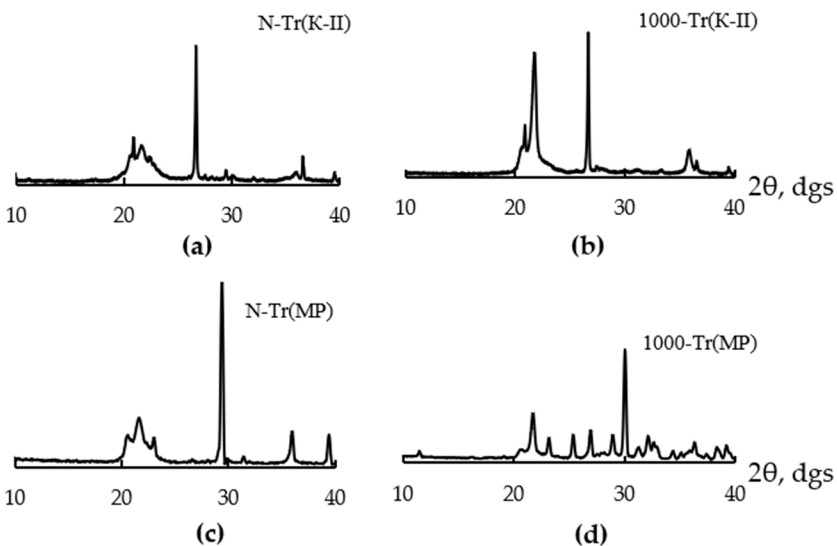

**Figure 1.** XRD patterns for natural (**a**,**c**) and thermally modified (**b**,**d**) tripoli samples.

**Table 1.** Phase compositions, phase contents, and crystallite sizes of natural and modified tripolis.

| Sample | Phases | Phase Content, wt % | D, nm | Sample | Phases | Phase Content, wt % | D, nm |
|---|---|---|---|---|---|---|---|
| N-Tr(K-I) | β-crist | 30.8 | 11 | H₂O-Tr(K-II) | β-crist | 42.1 | 8 |
|  | α-trid | 22.2 | 11 |  | α-trid | 29.9 | 6 |
|  | α-quartz | 31.9 | 248 |  | α-quartz | 20.3 | 248 |
| N-Tr(K-II) | β-crist | 36.0 | 9 | 3H-Tr(K-II) | β-crist | 44.7 | 8 |
|  | α-trid | 29.2 | 48 |  | α-trid | 25.7 | 7 |
|  | α-quartz | 23.4 | 126 |  | α-quartz | 23.8 | 248 |
|  | calcite | 47.3 | 555 | 1000-Tr(K-II) | β-crist | 19.6 | 21 |
| N-Tr(MP) | β-crist | 35.5 | 8 |  | α-trid | 39.7 | 13 |
|  | α-trid | 16.2 | 26 |  | α-quartz | 25.9 | >1000 |
|  | α-quartz | 0.6 | - |  | calcite | 49 | 93 |
| H₂O-Tr(K-I) | β-crist | 37.8 | 9 | H₂O-Tr(MP) | β-crist | 23 | 21 |
|  | α-trid | 55.6 | 18 |  | α-trid | 27 | 20 |
|  | α-quartz | 4.24 | 103 |  | α-quartz | 0.8 | >1000 |
| 3H-Tr (K-I) | β-crist | 20.1 | 9 |  | calcite | 44 | 93 |
|  | α-trid | 20.6 | 18 | 3H-Tr(MP) | β-crist | 31.3 | 21 |
|  | α-quartz | 50.0 | 292 |  | α-trid | 23.2 | 20 |
| 1000-Tr (K-I) | β-crist | 49.1 | 19 |  | α-quartz | 1.5 | >1000 |
|  | α-trid | 20.4 | 20 | 1000-Tr(MP) | wollast | 45.2 | 51 |
|  | α-quartz | 21.1 | 800 |  | β-larnite | 24.9 | 34 |
|  |  |  |  |  | α-crist | 12.2 | 6 |
|  |  |  |  |  | α-trid | 15.2 | 8 |

The contents of α-tridimite, α-quartz and β-cristobalite in N-Tr(K-I) and N-Tr(K-II) are unequal, where the α-tridimite and β-cristobalite contents in N-Tr(MP), which belong to carbonate tripolis, are high and the α-quartz content is very low (0.6 wt %).

The crystallite sizes of each phase in the natural tripolis that were studied was different: For β-cristobalite, they were between 8 and 11 nm, for α-tridimite, they changed from 11 to 48 nm, and for α-quartz, its crystallite size in N-Tr(K-I) was twice as much as that in N-Tr(K-II). The crystallite size of calcite in N-Tr(MP) was more than twice as much as the crystallite size of α-quartz in N-Tr(K-I).

As can be seen from Figure 1 and Table 1, the most significant changes for the modified tripoli samples, as compared with natural tripoli, take place in the samples calcined at 1000 °C. The reason for the alteration observed in both 1000-Tr(K) samples lies in the increase in both the α-tridimite contents and the crystallite sizes of their main phases (especially α-quartz). As opposed to 1000-Tr(K) samples, the α-tridimite content in 1000-Tr(MP) decreases and the calcite is transformed into two new

phases: Wollastonite ($CaSiO_3$) and β-larnite ($Ca_2SiO_4$), and α-cristobalite phase is generated instead of β-cristobalite.

Modification of N-Tr(K-I) and N-Tr(K-II), both in boiled water and in boiled acid, results in a significant increase in their β-cristobalite contents. For N-Tr(MP) modified by the same procedures, in addition to the alteration of the β-cristobalite and the α-tridimite phase ratio, the change in the calcite content takes place, especially in its crystallite size (from 555 down to 93 nm).

### 3.2. Water Vapor Sorption

Isotherms of water vapor adsorption-desorption by natural and modified tripolis are shown in Figure 2.

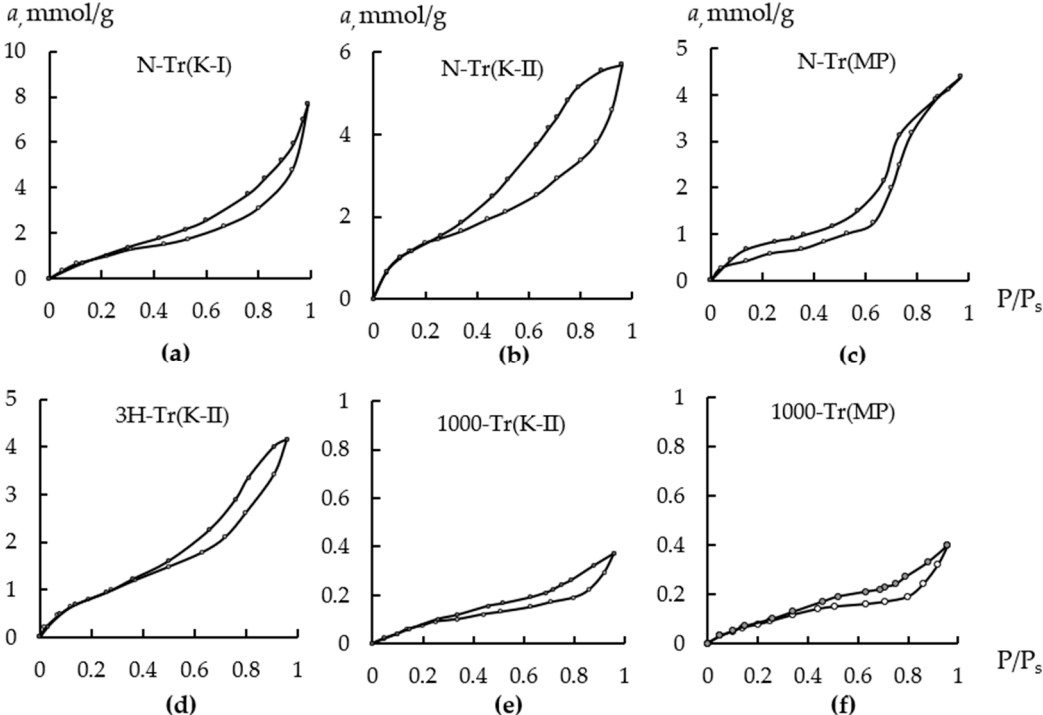

**Figure 2.** Isotherms of water vapor adsorption (○) and desorption (●) by some samples of natural (**a–c**) and modified (**d–f**) tripolis.

According to the well-known classification [24], the adsorption isotherms are assigned to Type IV. The capillary condensation hysteresis loops extend in the $P/P_s$ range from 0.3 to 0.98, which is characteristic of natural silicas [19], and belong to Type H3, whereas the first overlapping point of the N-Tr(MP) hysteresis loop is at $P/P_s \approx 0.01$, therefore determining its type is difficult. For the isotherms shown in Figure 2, evident alterations of their profiles as compared with natural tripolis take place: The hysteresis loop of 3H-Tr(K-II) becomes narrower and closes at $P/P_s = 0.4$, indicating some change in its pore structure. Water vapor adsorption drastically decreases for 1000-Tr.

The water vapor adsorption isotherms were analyzed using the linearized BET equation for polymolecular adsorption [24], and $P/P_s$ ranges from 0.05 to 0.35 were realized with a correlation coefficient ($R^2$) of 0.98–0.99 for all tripoli samples. Figure 3 shows the initial straight portions of the isotherms for the natural tripoli that was studied.

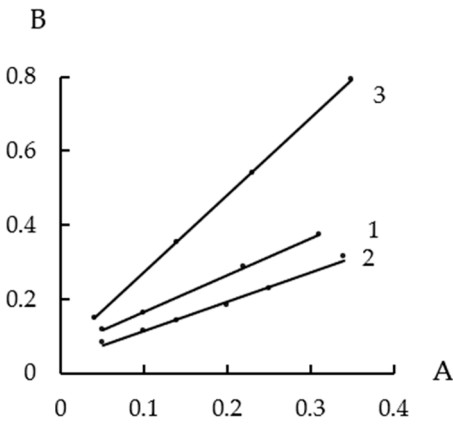

**Figure 3.** Initial linear portions of the water vapor adsorption isotherms in the BET equation coordinates ($A = P/P_s$; $B = A/a\,(1 - A)$) for natural tripolis studied in the current work: N-Tr(K-I) (1), N-Tr(K-II) (2), and N-Tr(MP) (3).

The parameters of the BET equation, i.e., the water monolayer capacities ($a_m$), the BET constants (C) and the specific surface areas ($S_{sp}$) are summarized in Table 2. Based on the desorption branches of the water vapor isotherms and using the Kelvin equation [24], pore diameter distribution curves were obtained and pore diameters equal to their maximum values are presented in Table 2. All samples except for N-Tr(K-I) and 3H-Tr(K-II) have the non-uniform mesoporous structures with pore diameters ranging from 2.0 to 13.7 nm, correlating with the earlier reported data [19].

**Table 2.** Structural-adsorption parameters of natural and modified tripolis.

| Sample | $a_m$, mmol/g | C | $a_\infty$, mmol/g | $S_{sp}$, m²/g | Maximum Values of Pore Diameter Distribution Curves, nm |
|---|---|---|---|---|---|
| N-Tr(K-I) | 0.92 | 16.71 | 7.7 | 60 | 5.2 |
| N-Tr(K-II) | 1.29 | 18.26 | 5.7 | 84 | 3.0, 5.5, 9.7 |
| N-Tr(MP) | 0.47 | 29.88 | 4.4 | 30 | 3.4, 8.5 |
| H₂O-Tr(K-I) | 1.14 | 6.02 | 7.7 | 74 | 2.5, 4.6, 8.0 |
| 3H-Tr(K-I) | 0.93 | 6.62 | 6.8 | 60 | 3.9, 7.0 |
| 1000-Tr(K-I) | 0.21 | 3.35 | 0.36 | 13 | 4.3, 9.4 |
| H₂O-Tr(K-II) | 0.84 | 25.70 | 5.0 | 54 | 1.9, 5.3, 8.6 |
| 3H-Tr(K-II) | 0.96 | 13.37 | 4.1 | 62 | 3.7 |
| 1000-Tr(K-II) | 0.13 | 3.39 | 0.37 | 8 | 4.4, 8.4, 13.7 |
| H₂O-Tr(MP) | 0.46 | 19.06 | 4.4 | 30 | 3.3, 8.4 |
| 3H-Tr(MP) | 0.80 | 17.30 | 5.10 | 52 | 3.3, 4.7, 8.4 |
| 1000-Tr(MP) | 0.09 | 8.59 | 0.40 | 6 | 9.5, 13.7 |

For the natural tripolis that were studied, $S_{sp}$ values decreased from 84 m²/g to 20 m²/g in the order of N-Tr(K-II), then N-Tr(K-I), followed by N-Tr(MP). Additionally, the α-tridimite contents in these tripolis decreased in a similar order. The β-cristobalite contents fluctuated with a maximum deviation of 5 wt %, and the α-quartz contents changed irregularly.

It is known that natural materials [25], metal oxides [26,27] and nanoparticles of metals and alloys [28] change in their composition, form and size of their particles after their treatment with water or acids. The adsorption properties of tripolis also differentially change as a result of their treatment with boiling water or boiling acid. Comparison of the $S_{sp}$ values for modified and natural tripolis showed that, in the former case, H₂O-Tr(K-I) > N-Tr(K-I), H₂O-Tr(K-II) < N-Tr(K-II) and H₂O-Tr(MP) ≈ N-Tr(MP), while in the latter case 3H-Tr(K-I) ≈ N-Tr(K-I), 3H-Tr(K-II) < N-Tr(K-II) and 3H-Tr(MP) > N-Tr(MP). Moreover, water adsorbability changes differentially as a result of the thermal treatment of various materials. For instance, the natural clinoptilolite calcination over a temperature range

of 400–1000 °C results in a decrease both in $S_{sp}$ and in its affinity to water molecules (the calcined sample becomes more hydrophobic) [25]. The adsorption capacities for silica gel after calcination over a temperature ranging from 200–1000 °C [24] and for plutonium dioxide calcined at 870 °C [29] change in a similar way. On the other hand, surface dehydroxylation in the case of Al, Cr, Fe, Ti, Zn and Mg oxides causes some increase in their affinity to water molecules. According to our data, tripolis calcined at 1000 °C demonstrate a 10–20 fold decrease in water vapor adsorption as compared with their natural counterparts.

The results obtained in the current work are evidence of changes in the structural-adsorption parameters of tripolis as a result of their modification. Their treatment in boiling water or acid causes apparent alterations in the contents of their main phases and sizes of their crystallites: The sizes of β-cristobalite and α-tridimite crystallites become similar, the size of α-quartz crystallites increases and the size of calcite crystallites substantially decreases (Table 1).

Calcination of N-Tr(K-I) and N-Tr(K-II) causes agglomeration of β-cristobalite, more substantial agglomeration of α-quartz and dehydroxylation of their surfaces resulted in a decrease in both $S_{sp}$ and the water adsorbability of these sorbents as in the case of silica gel [24]. In the case of 1000-Tr(MP), some decrease in its water adsorbability, in addition to the dehydroxylation of its surface, is due to the phase transformation of calcite into wollastonite ($CaSiO_3$) and larnite ($Ca_2SiO_4$) as well as the β-cristobalite transition into α-cristobalite. Calcite transformation to wollastonite was also reported elsewhere [30].

## 4. Conclusions

XRD analysis showed that tripolis of different origins have different phase compositions. Two tripolis from the Konoplianskyy deposit, N-Tr(K-I) and N-Tr(K-II), contained α-tridimite, β-cristobalite and α-quartz as their main phases. The calcite contents in them did not exceed 4.3 wt %. The calcite content in tripoli from the Mohyliv-Podils'kyy deposit, N-Tr(MP), belonging to carbonate tripolis, was 47.3%. The other main components of N-Tr(MP) were α-tridimite and β-cristobalite. Natural and modified tripolis substantially differ in the crystallite sizes of their main phases. All modification methods used in this work resulted in a change in the tripoli phase contents, even in phase transformation in the case of N-Tr(MP) which was calcinated at 1000 °C.

The isotherms of water vapor adsorption by natural and modified tripolis assigned as Type IV have been analyzed using the BET equation. Some correlations between the adsorption parameters of tripolis and their phase compositions, main phase contents and crystallite sizes have been found. By analyzing the desorption branches of the water vapor isotherms, it can be asserted that the natural and modified tripolis studied in the current work have non-uniform mesoporous structures with various pore diameters ranging from 2.0 to 13.7 nm.

**Author Contributions:** All the authors conceived the study, formulated the research idea, made the calculations and prepared the manuscript draft version.

**Funding:** The study was carried out with the support of the Ministry of Education and Science of Ukraine.

**Conflicts of Interest:** The authors declare no conflict of interest.

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
