# Peer review of "Effect of Both the Phase Composition and Modification Methods on Structural-Adsorption Parameters of Dispersed Silicas"

_colloids, doi:10.3390/colloids3010001_

Round 1
Reviewer 1 Report
The authors have studied phase compositions of tripolis from two Ukrainian deposits in their natural and differently modified forms. Water vapor adsorption-desorption isotherms onto the natural and differentially modified tripolis have also been investigated. The work is interesting and the conclusions are justified. This reviewer suggests the authors to make the following amendments in the revised version of their manuscript.
Provide SEM picture of both natural and differentially modified Ukrainian tripolis.
Provide an experimental uncertainty analysis.
The introduction section does not provide sufficient background and needs to cite the following relevant recent papers.
B. Choudhury et al., An overview of developments in adsorption refrigeration systems towards a sustainable way of cooling, Applied Energy, Vol. 104, pp. 554-567, 2013.
M. Younes, I.I. El-Sharkawy, A.E. Kabeel, et al., A review on adsorbent-adsorbate pairs for cooling applications, Applied Thermal Engineering, Vol. 114 pp. 394-414, 2017.
B.B. Saha, I.I. El-Sharkawy, M.W. Shahzad, et al., Fundamental and application aspects of adsorption cooling and desalination, Applied Thermal Engineering Vol. 97 Special Issue: SI pp. 68-76, 2016
A diligent proof reading is recommended.
Author Response
Response to Reviewer 1 Comments
Point 1: Provide SEM picture of both natural and differentially modified Ukrainian tripolis.
Response 1: Scanning electron microscopy investigations of natural and modified tripoli samples are carried out in cooperation with Institute of Monocrystals of National Academy of Sciences ofUkraine. To the moment, our investigations are not completed: we have only few SEM images (some examples are shown in the attached file) and their description. We plan to obtain more information and present our results in next paper.
Point 2: Provide an experimental uncertainty analysis.
Response 2: An error of measurement was ± 2 %.
Point 3: The introduction section does not provide sufficient background and needs to cite the following relevant recent papers.
Choudhury, B.; Saha, B.B.; Chatterjee,P.K.; Sarkar, J.P. An overview of developments in adsorption refrigeration systems towards a sustainable way of cooling. Appl. Energy, 2013, 104, 554–567.
Younes, M.M.; El-Sharkawy, I.I.; Kabeel, A.; Saha, B.B. A review on adsorbent-adsorbate pairs for cooling applications. Appl. Therm. Eng., 2017, 114, 394–414.
Saha, B.B.; El-Sharkawy, I.I.; Shahzad, M.W.; Thu, K.; Ang, L.; Ng, K.C. Fundamental and application aspects of adsorption cooling and desalination, Appl. Therm. Eng., 2016, 97, 68–76.
Response 3: We revised the Introduction and added the References with these items.

Reviewer 2 Report
Manuscript ID: colloids-406650
This paper presents the effect of the phase composition and modification methods on the structural and adsorption parameters of Tripolis from different Ukrainian deposits. Such investigations are important because of wide applications of such materials in different branches of industry and technology. However some remarks concerning the preparation of the manuscript should be taken into consideration while revising the paper:
What are the differences between N-Tr(K-I) and N-Tr(K-II) samples origin?
line 59: enclosedvolume please indicate space
line 61: what 21 _C means?
line 71: however. slightly amorphized – please correct point to comma
line 77: please explain: (d = 3.039 A) - this abbreviation was not discussed in the text
Table 1: please change commas to points
line 85: tridimite. -quartz, - please correct point to comma
In my opinion, the two last paragraphs of the Results (lines from 149 to 160) should be included in the Conclusions section.
Author Response
Response to Reviewer 2 Comments
Point 1: What are the differences between N-Tr(K-I) and N-Tr(K-II) samples origin?
Response 1: The samples were mined from different depths of occurrence.
Point 2: line 59: enclosedvolume please indicate space
line 61: what 21 _C means?
line 71: however. slightly amorphized – please correct point to comma
line 77: please explain: (d = 3.039 A) - this abbreviation was not discussed in the text
Table 1: please change commas to points
line 85: tridimite. -quartz, - please correct point to comma
Response 2: All punctuation errors were corrected.
Point 3: In my opinion, the two last paragraphs of the Results (lines from 149 to 160) should be included in the Conclusions section.
Response 3: We did not include two last paragraphs of the Results in the Conclusions because they contain result discussion with the involvement of some data reported in literature ([24] and [30]).